# fMRI Brain Decoding and Its Applications in Brain–Computer Interface: A Survey

**DOI:** 10.3390/brainsci12020228

**Published:** 2022-02-07

**Authors:** Bing Du, Xiaomu Cheng, Yiping Duan, Huansheng Ning

**Affiliations:** 1School of Computer and Communication Engineering, University of Science and Technology Beijing, Beijing 100083, China; dubing@ustb.edu.cn (B.D.); s20190669@xs.ustb.edu.cn (X.C.); 2Department of Electronic Engineering, Tsinghua University, Beijing 100084, China; yipingduan@mail.tsinghua.edu.cn

**Keywords:** brain decoding, variational autoencoder (VAE), generative adversarial network (GAN), graph convolutional networks (GCN), functional magnetic resonance imaging (fMRI), brain–computer interface (BCI)

## Abstract

Brain neural activity decoding is an important branch of neuroscience research and a key technology for the brain–computer interface (BCI). Researchers initially developed simple linear models and machine learning algorithms to classify and recognize brain activities. With the great success of deep learning on image recognition and generation, deep neural networks (DNN) have been engaged in reconstructing visual stimuli from human brain activity via functional magnetic resonance imaging (fMRI). In this paper, we reviewed the brain activity decoding models based on machine learning and deep learning algorithms. Specifically, we focused on current brain activity decoding models with high attention: variational auto-encoder (VAE), generative confrontation network (GAN), and the graph convolutional network (GCN). Furthermore, brain neural-activity-decoding-enabled fMRI-based BCI applications in mental and psychological disease treatment are presented to illustrate the positive correlation between brain decoding and BCI. Finally, existing challenges and future research directions are addressed.

## 1. Introduction

In recent years, the concept of the brain–computer interface has gradually entered the public’s field of vision and has become a hot topic in the field of brain research. Brain neural activity decoding is a key technology for the brain–computer interface. Therefore, this paper first surveys research in the field of visually decoding brain neuronal activity, explains the strengths and weaknesses of these studies, and updates recent research progress and potential clinical applications of brain–computer interfaces in psychotherapy. Finally, potential solutions are proposed for the problems existing in the current brain activity decoding models. Functional magnetic resonance imaging (fMRI) is a new neuroimaging method. Its principle is to use magnetic resonance imaging to measure the changes in hemodynamics caused by neuronal activity. From the perspective of neuroscience and neuroimaging, functional magnetic resonance imaging can be used to decode the perception and semantic information of the cerebral cortex in a non-invasive manner [1]. The spatial resolution of fMRI is high enough, while the noise contained in the measurement of the brain activity is relatively small [2]. The real-time fMRI monitors the activity state of the cerebral cortex through the blood-oxygen-level-dependent (BOLD) variation induced by brain neural activity and simultaneously collects and analyzes the BOLD signals of the brain [3,4]. In recent years, more and more researchers have adopted real-time fMRI to investigate the brain’s self-regulation and the connection among different brain regions [5,6,7]. Compared with Electroencephalogram (EEG) [2,8,9,10,11], fMRI has higher spatial resolution and suffers from lower noise. Therefore, decoding brain neural activity with fMRI data has become more popular [12,13,14,15,16]. Contrary to the visual encoding model [17,18,19,20], the decoding model predicts the visual stimuli through the neural responses in the brain. The current research on the decoding of visual stimuli can be divided into three categories: classification, recognition, and reconstruction.

The difficulty in brain decoding is the reconstruction of visual stimuli through the cognition model of the brain’s visual cortex, as well as learning algorithms [21,22,23,24]. The initial research on reconstructing visual stimuli used retinal shadows [25,26] or fMRI [12] to observe the response of the visual cortex and map a given visual stimuli to the brain neural activity linearly. These studies considered the reconstruction of the visual stimulus as a simple linear mapping problem between the monomer voxels and brain activities. Then, the researchers further used multi-voxel pattern analysis (MVPA) combined with the machine learning classification algorithms to decode the information from a large number of voxel BOLD signals [14,16,27,28,29]. For example, multi-voxel patterns were used to extract multi-scale information from fMRI signals to achieve a better performance of the reconstruction [29]. According to the sparsity of the human brain’s processing of external stimuli, ref. [14] added sparsity constraints to the multivariate analysis model of Bayesian networks to quantify the uncertainty of voxel features [16]. Because of the diversity of natural scenes and the limitations of recording neuronal activities in the brain, some studies try to include the a priori of natural images in multiple analysis models to reconstruct some simple images [23,30,31]. Using two different coding models to integrate the information of different visual areas in the cerebral cortex and based on a large number of image priors, ref. [31] reconstructed natural images from fMRI signals for the first time. In addition, the researchers also used machine learning [32] to decode the semantic information of human dreams [33] and facial images [34]. In order to trace the reversible mapping between visual images and corresponding brain activity, ref. [22] developed a Bayesian canonical correlation analysis (BCCA) model. However, its linear structure makes the model unable to present multi-level visual features, and its spherical covariance assumption cannot meet the correlation between fMRI voxels [21]. Ref. [24] introduced a Gaussian mixture model to perform percept decoding with the prior distribution of the image, which can infer high-level semantic categories from low-level image features by combining the prior distribution of different information sources [24]. Moreover, ref. [35] explored the impact of brain activity decoding in different cerebral cortex areas. Researchers believe that the high-level visual cortex, i.e., the ventral temporal cortex, is the key to decoding the semantic information and natural images from brain activity [31].

Significant achievements have been made in image restoration and image super-resolution with deep learning algorithms. The structure of deep neural networks (DNN) is similar to the feedforward of the human visual system [36], so it is not surprising that DNN can be used to decode the visual stimulus of the brain activity [30,37,38] mapped multi-level features of the human visual cortex to the hierarchical features of a pre-trained DNN, which can make use of the information from hierarchical visual features. Ref. [37] combined deep learning and probabilistic reasoning to reconstruct human facial images through nonlinear transformation from perceived stimuli to latent features with the adversarial training of convolutional neural networks.

Therefore, this paper surveys the deep neural networks used in visual stimulus decoding, including Convolutional Neural Networks (CNN), Recurrent Neural Networks (RNN), and Graph Convolutional Neural Networks (GCN). Deep neural networks, especially deep convolutional neural networks, have proven to be a powerful way to learn high-level and intermediate abstractions from low-level raw data [39]. Researchers used pre-trained CNN to extract features from fMRI data in the convolutional layer [28,38,40]. Furthermore, ref. [41] adopted an end-to-end approach and trained the generator directly using fMRI data. The feature extraction process of RNN is similar to that of CNN, but the process is not affected by the input time sequence [30]. The long short-term memory model (LSTM) is a typical RNN structure and is also used to decode brain activity [42,43,44]. However, RNN only considers the time series of the BOLD signal and ignores spatial correlations between different functional areas of the brain [30]. GCN can investigate the topological structure of the brain functional areas and then predict the cognitive state of the brain [45,46]. With limited fMRI and labeled image data, the GCN-based decoding model can provide an automated tool to derive the cognitive state of the brain [47].

Owing to the rapid adoption of deep generative models, especially Variational Auto-encoders (VAEs) and Generative Adversarial Networks (GANs), a large number of researchers have been encouraged to study non-natural image generation through VAEs and GANs [2,48].

VAE has a theoretically backed framework and is easy to train. Traditional methods usually separated encoding and decoding processes. The authors of [49] thought that the encoding and decoding models should not be mutually exclusive, and the prediction can be more accurate through an effective unified encoding and decoding framework. VAE is such a two-way model, which composes of an encoder, a decoder, and latent variables in between. VAE is trained to minimize the reconstruction error between the encoded–decoded data and the initial data [19,23,38]. Instead of taking an input as a single point, VAE takes the input as a distribution over the latent space [50] and naturally represents the latent space regularization. The encoder learns latent variables from the input, and the decoder generates an output based on samples of the latent variables. Given sufficient training data, the encoder and the decoder are trainable altogether by minimizing the reconstruction loss and the Kullback–Leibler (KL) divergence between the distributions of latent variables and independent normal distributions. However, the synthesized image derived from VAE lacks details [48]. At present, some researchers have tried to reconstruct the visual stimuli from the associated brain activities based on VAE [21,38,48,51].

GAN is widely used for the images synthesization [52,53] and image-to-image conversions [54,55]. The advantage of GAN is that the Markov chain is no longer needed, and the gradient can be obtained by back propagation. Various factors and interactions can be incorporated into the model during the learning process without inference [56]. GAN can synthesize relatively clear images, but the stability of training and the diversity of sampling are its major problems [48]. In brain decoding, researchers have successfully applied GAN-based models to decode natural images from human brain activity [57,58,59,60,61]. Although GAN can be used to generate very good images, the quality and variety of the images reconstructed from brain activity data are quite limited [62]. The significance of the generative models lies in the representation and manipulation of the high-dimensional probability distributions [63]. The generative model can be trained with lossy data (some samples are unlabeled) and be performed by semi-supervised learning, which reduces the difficulties of sampling the brain data [63].

Studying brain decoding intends to create brain–computer interfaces, which uses the brain neuron signals to control the device in real time [30], such as controlling the robotic arm or the voice generator. Another goal of studying brain neuron decoding is to better understand the cognitive structure of the brain by constructing a decoding model [57]. For people with mental illness, neuro-feedback therapy can help patients restore healthy cognition. The fMRI brain–computer interface (fMRI-BCI) collects signals in a single or multiple regions of interest (ROI) in the brain; researchers used a linear discriminant classifier [64] and a multi-layer neural network [14] to classify the scanned fMRI signals. Since the differences in mental activity and physical structure between people results in differences in the number and location of the active area of each person’s brain’s ROI, multivariate analysis is more appropriate for decoding brain activity [13,29,65]. The analysis of multi-voxel patterns and the development of neuro-imaging technology can be applied to the physical examination and diagnosis and rehabilitation training [13,65]. In order to decode the spiritual and psychological information in fMRI, ref. [66] used the Support Vector Machines (SVM) to classify and identify a variety of different emotional states. The participants’ brain state decoded from fMRI signals can be regarded as a BCI control signal, which provides the neuro-feedback to the participants [65]. However, many technical challenges still remain in fMRI brain decoding, i.e., the noise included in fMRI signals and the limited high-resolution fMRI signal data set [67].

After determining the topic of this survey, we first conduct a literature search. The database for the literature search is the Web of Science Core Collection, the keywords are fMRI, Brain decoding, BCI, and the time span is 1995 to 2020. Then, we performed literature screening. We imported the literature information retrieved in the previous step into CiteSpace (data mining software) for clustering analysis, and then, according to the high-frequency nodes (representing highly cited literature) in the clustering results and high betweenness centrality nodes (representing documents that form a co-citation relationship with multiple documents), we screened out 120 articles that met the requirements.

The rest of the paper is organized as follows. Section 2 introduces the basics of the brain activity decoding model. Section 3 introduces the hot research interests and specific examples in this area. Section 4 introduces the specific medical applications of fMRI-BCI. Section 5 analyzes the current challenges in using fMRI to decode brain activity and possible solutions. Section 6 is a summary of the survey.

## 2. Brain Decoding Based on Learning

Visual pathway is a typical illustration of brain encoding and decoding with regard to visual stimuli. Visual pathway processes visual information after the retina receiving a visual stimuli [68]. The optic nerve is responsible for transmitting the special sensory information for vision. The optic nerves from each eye unite to form the optic chiasm, then to the lateral geniculate body, which projects the visual information to the primary visual cortex. Figure 1 illustrates how the visual cortex of brain encodes the visual information and reflected by an fMRI. Additionally, the decoding process is the inverse of the encoding process. In this section, we elaborate fMRI brain decoding models based on machine learning.

### 2.1. The Relationship between Brain Encoding and Decoding

The encoding and decoding of brain neuron activity are the two fundamental aspects of human’s cognitive visual processing system [15,69]. The encoding process predicts brain activities based on visual stimuli aroused by the external environment [21]. Contrary to the encoding process, the decoding process analyzes brain activities to retrieve the associated visual stimuli [21]. The encoding process describes how to obtain information in voxels corresponding to one region, and once the encoding model is constructed, one can derive the decoding model using Bayesian derivation [15]. Moreover, decoding model can be applied to verify the integrity of encoding process [15]. The encoding model can be trained to predict the representations of neuronal activity, which in turn helps to extract features from the decoding model, thereby improving the decoding performance of brain activities [70]. Therefore, encoding and decoding models are not mutually exclusive because the prediction can be performed more accurately by an effective unified encoding and decoding framework. Theoretically, the general linear model (GLM) can be used to predict the voxel activity, so GLM can be regarded as an encoding model [49]. The review papers that survey brain activity encoding models are listed in Table 1.

### 2.2. Machine Learning and Deep Learning Preliminaries

We briefly describe the traditional machine learning and deep learning methods used in brain decoding models.

#### 2.2.1. Machine Learning

The mapping of activities in the brain to cognitive states can be treated as a pattern recognition problem. In this regard, the fMRI data are a spatial pattern, and thus a statistical pattern recognition method (e.g., a machine learning algorithm) can be used to map these spatial patterns to instantaneous brain states. The machine learning algorithms are used to learn and classify multivariate data based on the statistical properties of the fMRI data set. Surveys on brain decoding models based on machine learning are listed in Table 2. In the surveyed literature, we summarized the traditional machine learning methods applied to brain decoding as follows:

(1) **Support vector machine (SVM)** is a linear classifier, which aims to find the largest margin hyperplane to classify the data [64]. The distance between the data of different categories is maximized through maximal margin. SVM can be extended to act as a nonlinear classifier by employing different kernel functions. However, fMRI data are highly dimensional with few data points, hence a linear kernel is sufficient to solve the problem.

(2) **Naive Bayesian classifier (NBC)** is a probabilistic classifier, which is based on Bayes theorem and strong independent assumptions between features. NBC is widely used in brain decoding models. When processing fMRI data, NBC assumes that each voxel’s fMRI is independent. For instance, the variance/covariance of the neuronal and hemodynamic parameters are estimated on a voxel-by-voxel basis using a Bayesian estimation algorithm, enabling population receptive fields (pRF) to be plotted while properly representing uncertainty about pRF size and location based on fMRI data [72]. In addition, NBC is often combined with other methods, such as Gaussian mixture models and convolutional neural networks.

(3) **Principal Component Analysis (PCA)** maps data to a new coordinate system through orthogonal linear transformation [73]. When processing high-dimensional fMRI data, PCA is used as a dimensionality reduction tool to remove noise and irrelevant features, thereby improving the data processing speed. In addition, the risk of over-fitting is avoided due to the reduction in training data in order to improve the generalization ability of the model [40].

#### 2.2.2. Deep Learning

The hierarchical structure of DNN is similar to the feed-forward of the human visual system, and DNN can be used to encode external stimuli [36]. In particular, DNN is used to explore how the brain maps the complex external stimulus to corresponding areas in the cerebral cortex, which quantitatively shows that an explicit gradient for feature complexity exists in the ventral pathway of the human brain [74]. In the surveyed literature, DNN models are summarized as follows:

(1) **Recurrent Neural Network (RNN)** is a generalization of a feedforward neural network that has an internal memory. The hidden nodes of a RNN form a cyclic structure, and the state of the hidden layer at each stage depends on its past states. This structure allows RNN to save, memorize and process complex signals in the past for a long time. RNN can process variable length sequences of inputs, which occurs in neuroscience data [30]. The main advantage of using RNNs instead of standard neural networks is that the weights are shared between neurons in hidden layers across time. Therefore, RNN can process the previously input sequence, as well as the upcoming sequence to explore the temporal dynamic behavior. Therefore, RNNs can be applied to decode brain activity since they can process correlated data across time [57].

(2) **Convolutional Neural Network (CNN)** consists of three parts: convolution layers, pooling layers, and fully connected layers. CNNs can handle many different input and output formats: 1D sequences, 2D images, and 3D volumes. The most important task of the CNN is to iteratively calibrate the network weights through the training data, also known as the back propagation algorithm. For decoding different neurons activities, it can be realized via weights sharing across convolutional layers for learning a shared set of data-driven features [30]. Moreover, the generative models VAE and GAN based on CNN have attracted much attention recently. The deep generative models for brain decoding are listed in Table 3.

(3) **Graph Convolutional Network (GCN)**: At present, most deep-learning-based brain decoding research has not considered the function correlations and dynamic temporal information between human brain regions [45]. For brain decoding, GCN can consider the connections among brain regions when performing brain decoding via fMRI [77]. Unlike CNN, GCN extracts features effectively, even with completely random initialized parameters [78]. Ref. [46] extracted fMRI data features with GCN to predict brain states. Given labeled data, compared to the other classifiers, GCN performs better on a limited data set. In this survey, GCN-based brain decoding models are listed in Table 4.

## 3. Brain Decoding Based on Deep Learning

In this section, the principles and specific applications of VAE, GAN, and GCN that have received the most attention in brain decoding are introduced.

### 3.1. VAE-Based Brain Decoding

VAE is a generative model that contains hidden variables, and it combines variational Bayesian estimation with a neural network, which learns variational deduction parameters and thus obtains the optimized inference through maximum likelihood (ML) or Max a posteriori (MAP) [32,79]. Let 
x
 represent the input image and 
z
 represent the latent variables. Latent variables can be considered as dependencies between features in multiple dimensions. Latent variables reconstruct the data 
x
 through a generative network, while generates new data that do not exist in the original data. For the image data set, latent variables 
z
 are the implicit factors that determine the features of an image 
x
. The inference network in VAE infers hidden variables 
z
. The process of variational inference is also a process of dimensionality reduction [79]. The generative network in VAE uses the randomly sampled latent variables 
z
 to generate an image 
x
 [32].


ϕ
 is an inferred model that *x* infers *z*; 
θ
 is a generative model where *z* generates *x*. The marginal likelihood of an input image *x* can be written as:
(1)
$$logp_{\theta }\left(x\right)=D_{KL}\left[q{}_{\phi }\left(z|x\right)∥p{}_{\theta }\left(x|z\right)\right]+L(\theta ,\phi ;x).$$


Given *x*, the probability that 
ϕ
 infers *z* is 
$q_{\phi }\left(z|x\right)$
; given *z*, the probability that 
θ
 generates *x* is 
$p_{\theta }\left(x|z\right)$
. The first term is a non-negative Kullback–Leibler (KL) difference, so 
L(θ,ϕ;x)
 can be considered as the lower bound of Equation (Equation 1). The learning rules for VAE are to maximize Equation (Equation 1), which is equivalent to maximizing 
L(θ,ϕ;x)
, as shown in Equation (Equation 2):
(2)
$$\begin{array}{cc} L(\theta ,\phi ;x) & =-D_{KL}\left[q_{\phi }\left(z|x\right)∥p_{\theta }\left(z\right)\right]\\ \; & +E_{z∼q_{\phi }\left(z|x\right)}\left[\log\left(p_{\theta }\left(x|z\right)\right)\right] \end{array}$$


In Equation (Equation 2), the first term is the Kullback–Leibler (KL) difference between the distribution of latent variable *z* inferred from *x* and the prior distribution of *z*; the second term is the expectation of the log-likelihood that the input image *x* is generated by *z* sampled from the inferred distribution 
$q_{\phi }\left(z|x\right)$
. If 
$q_{\phi }\left(z|x\right)$
 follows a multivariate distribution with unknown expectation 
μ
 and variance 
σ
, the objective function is differentiable with respect to 
(θ,ϕ,μ,σ)
 [32]. The parameters of VAE can be optimized by stochastic gradient descent [75]. The brain learns from experience in an unsupervised way. Likewise, VAE uses deep neural networks to learn representations from large amounts of data in such an unsupervised way as the brain does [50]. Furthermore, VAE-based brain decoding models can be found in [50,51,67].

Du et al. [51] proposed a structured deep generative neural decoding model, which consists of a structured multiple-output regression model (SMR) and an introspective conditional generation (ICG) model, to reconstruct visual images from brain activity, as shown in Figure 2. This conditional deep generative model (DGM) combines the advantages of VAE and GAN to reconstruct high-quality images with stable training. Specifically, SMR calculates the correlation between fMRI voxels on the CNN units, as well as the correlation between the output of each linear model. Then, the ICG model is proposed to decode these correlated CNN features extracted by SMR, inspired by the IntroVAE [48].

The key point of ICG is the IntroVAE that modifies the LOSS function of VAE and allows the encoder to perform a discriminator’s function in a GAN without introducing a new network. The ICG model can estimate the difference between the generated image and the training data in an introspective way. Similar to the original VAE, the training of ICG is to optimize encoder 
ϕ
 and decoder 
θ
 iteratively until convergence:
(3)
$$\hat{\theta }=\arg\min_{\theta }\left[L_{AE}+\alpha D_{KL}\left(q_{\phi }\left(z|x_{f},h\right)∥p\left(z\right)\right)\right],$$


(4)
$$\hat{\phi }=\arg\min_{\phi }\left[L_{AE}+\beta D_{KL}\left(q_{\phi }\left(z|x,h\right)∥p\left(z\right)\right)-\alpha D_{KL}\left(q_{\phi }\left(z|x_{f},h\right)∥p\left(z\right)\right)\right].$$

where *h* represents the output CNN features from SMR, *x* represents the input training image, *z* represents the hidden variable, and 
z∼p(z)
 or 
$z∼q_{\phi }\left(z|x,h\right)$
. 
$x_{f}$
 is a fake image generated by 
$p_{\theta }\left(x|z,h\right)$
. 
$L_{AE}$
 is the error of VAE reconstruction. For real image data points, Equations (Equation 3) and (Equation 4) cooperate with each other rather than conflict with each other, which can be considered as a conditional variational autoencoder (CVAE) [80]. In this case, 
$D_{KL}(q_{\phi }\left(z\right|x_{f},h∥p\left(z\right))$
 in Equations (Equation 3) and (Equation 4) does not work. The encoder and **decoder cooperates to minimize the reconstruction loss**

$L_{AE}$
. Equation (Equation 4) regularizes the encoder by matching 
$q_{\phi }\left(z|x,h\right)$
 with 
p(z)
. For fake image data points, Equations (Equation 3) and (Equation 4) confront with each other, which can be considered as a conditional generative adversarial network (CGAN) [56]. From Equations (Equation 3) and (Equation 4), we can see that 
$D_{KL}\left(q_{\phi }\left(z|x_{f},h\right)∥p\left(z\right)\right)$
 are mutually exclusive during the training process. Specifically, Equation (Equation 3) hopes to match posterior 
$q_{\phi }\left(z|x_{f},h\right)$
 with prior distribution 
p(z)
 to minimize 
$D_{KL}\left(q_{\phi }\left(z|x_{f},h\right)∥p\left(z\right)\right)$
, while Equation (Equation 4) hopes to match posterior 
$q_{\phi }\left(z|x_{f},h\right)$
 with prior distribution 
p(z)
 to maximize 
$D_{KL}$
. Generally, Equation (Equation 3) (generative model) tends to generate a realistic image as much as possible so that Equation (Equation 4) (discrimination model) cannot discriminate its authenticity. The 
α
 and 
β
 in Equations (Equation 3) and (Equation 4) are parameters, which balance CVAE and CGAN.

### 3.2. GAN-Based Brain Decoding

GAN consists of a generative network and a discriminant network, and tends to generate high-quality pictures through adversarial training of the discriminator [81]. Specifically, the discriminator distinguishes whether the sample comes from the generated model or the probabilistic model of the training data. The generator is trained to generate realistic pictures for maximizing the error rate of the discriminator. The two models can theoretically achieve the Nash equilibrium (the probability out of the discriminator is 
0.5
) for creating a remarkably nature-like picture. Let *D* and *G* denote the discriminator and generator, respectively. The optimization objective function of GAN can be expressed by Equation (Equation 5):
(5)
$$\min_{G}\max_{D}\left(D,G\right)=E_{x∼p_{data}\left(x\right)}\left[\log D\left(x\right)\right]+E_{n∼p_{n}\left(n\right)}\left[\log\left(1-D\left(G\left(n\right)\right)\right)\right].$$


The distribution of the real image data *x* is 
$p_{data}\left(x\right)$
, and the distribution of the artificial noise variable *n* is 
$p_{n}\left(n\right)$
. The input variable of G is *n*, which outputs the distribution of the real data. The input of D is the real data *x*, which outputs a scalar 
D(x)
, a probability of the data being real. The objective function maximizes 
D(x)
 by training D and minimizes 
log(1−D(G(n))
 by training G. Because *G* is a differentiable function, the gradient-based optimization algorithm can be used to obtain a global optimal solution [82].

A shape-Semantic GAN is proposed in [76], which consists of a linear shape decoder, a semantic decoder based on DNN, and an image generator based on GAN. The fMRI signals are input into the shape decoder and semantic decoder. The shape decoder reconstructs the contour of the visual image from the lower visual cortex (V1, V2, V3), and the semantic decoder is responsible for extracting the semantic features of the visual image in the higher visual cortex (FFA, LOC, PPA). The output of the shape decoder is input to the GAN-based image generator, and the semantic features in GAN can compensate the details for the image to reconstruct high-quality images.

The linear shape decoder trains models for the activities of V1, V2, and V3 viewed from fMRI, and combines the predicted results linearly to reconstruct the shape of the natural image. The combined training model of the shape decoder is shown in Equation (Equation 6):
(6)
$$\begin{array}{c} r_{sp}\left(i,j\right)=\sum w_{ij}^{k}p_{k}^{*}\left(i,j\right),\;k=V1,V2,V3, \end{array}$$

where 
$w_{ij}^{k}$
 represents the weight of the image pixel at position 
(i,j)
, and 
$p_{k}^{*}\left(i,j\right)$
 represents the predicted image pixel at position 
(i,j)
. It can be seen from Equation (Equation 6) that the decoded image shape 
$r_{sp}$
 is a linearly combination of image pixels 
$p_{k}^{*}\left(i,j\right)$
. The semantic decoder, consisting of an input layer, two intermediate layers, and an output layer, is a lightweight DNN model. In the training phase, the high-level visual cortex activity recorded by fMRI is the input. The Tanh activation function is used to extract semantic features in the middle layer, and the sigmoid activation function is used to classify the images in the output layer.

In the final image generation stage, the original GAN can extract high-level semantic features. However, some low-level features (such as texture) may be lost in the process of encoding/decoding, which may cause the reconstructed image deformed and blurred. Therefore, U-Net [83] can be used as the generator, which consists of a pair of a symmetrical encoder and decoder. The U-Net breaks through the bottleneck of the structure. The U-Net does not require the low-level features necessarily to reconstruct the image, but it can extract the high-level features of the image. Moreover, the semantic features can be input to the U-Net decoder, which is equivalent to optimize a GAN generator with the constraints of the semantic features and shape features. In particular, the output of the shape decoder together with the output of the U-Net are input to the discriminator, which compares the high-frequency structures of the two and thus guides the generator training. Let 
$G_{s}$
 represent a U-Net generator, and 
$D_{t}$
 represent a discriminator. The conditional GAN model is shown in Equation (Equation 7):
(7)
$$\begin{array}{c} L\left(s,t\right)=L_{adv}\left(s,t\right)+\lambda_{img}L_{img}\left(s\right), \end{array}$$

where *s* and *t* represent the parameters of the generator and discriminator, respectively. 
$L_{adv}\left(s,t\right)$
 represents the adversarial loss, and 
$L_{img}\left(s\right)$
 and 
$\lambda_{img}$
 represent the image loss and weight, respectively. The conditional GAN discriminator is designed to model both low-frequency structures and high-frequency structures, thereby guiding the generator to synthesize images with more details.

### 3.3. Graph Convolutional Neural Networks

Unlike CNN and RNN, Graph Neural Networks (GNN) decode neuronal activity with brain connectivity [77]. Recently, some researchers have used Graph Convolutional Neural Networks (GCN) to predict [46] and annotate [47] brain consciousness and then predict the gender and age of the subjects [45]. When the training data set is quite limited, GCN shows better decoding performance than other classifiers.

GCN is first proposed in [78]. In the considered fMRI data, there are *N* nodes, and each node has *d*-dimensional features. The nodes’ features form an 
N×d
 matrix *X*. The relationships between the nodes form an 
N×N
 matrix *A*, known as an adjacency matrix. *X* and *A* are the inputs to the model. The propagation of GCN feature layers can be shown in Equation (Equation 8): 
(8)
$$f\left(H^{\left(l\right)},A\right)=\sigma \left({\hat{Q}}^{-\frac{1}{2}}\hat{A}{\hat{Q}}^{-\frac{1}{2}}H^{\left(l\right)}W^{\left(l\right)}\right),$$

where *l* is an identity matrix; *H* represents the feature matrix of each layer; *W* represents the initialized parameter matrix; and 
σ
 represents a nonlinear activation function. For the input layer, *X* is *H*. The diagonal elements of the relationship matrix *A* is 0, that is, when performing inner product with feature matrix *H*, the node’s own features will be ignored, which can be solved by 
$\hat{A}=A+I$
. The key of the GCN is the symmetric normalized Laplacian matrix 
${\hat{Q}}^{-\frac{1}{2}}\hat{A}{\hat{Q}}^{-\frac{1}{2}}$
 in Equation (Equation 8), where the degree matrix 
$\hat{Q}$
 of 
$\hat{A}$
 can be directly calculated. Therefore, when 
$\hat{A}$
 is multiplied by the feature matrix *H*, the distribution of the features will not be changed.

However, the methods based on GCN [47] and RNN [43,84] ignore the details of the fMRI signals and the dependence between different brain functional areas. Inspired by the work of spatio-temporal graph convolutional network (ST-GCN) developed in [85,86] to predict the graph structure, ref. [45] applied ST-GCN to brain decoding, where the temporal graph explores the dynamics brain activity, and the spatial graph explores the functional connectivity between different brain regions. The input of ST-GCN extracts d-dimensional spatio-temporal features 
u
 from *N* ROI regions of the brain, thus, 
$u\in R^{N\times d}$
. ST-GCN is composed of three spatio-temporal convolution (ST-GC) layers. The time kernel size of each ST-GC layer is 
Γ=11
; the step size is 1; and the drop rate is 0.5. Each ST-GC layer outputs 64 channels. An edge importance matrix 
$M\in R^{N\times N}$
 is added between ST-GC layers. Specifically, when a certain node is spatially convoluted, the contribution of its adjacent nodes will be redefined by the row of the edge importance matrix where the node is located. Then, the 64-dimensional feature vector is generated through global average pooling and fed into the fully connected layer. The sigmoid activation function is then used to activate before the loss function calculated the classification probability. After back propagation, the stochastic gradient descent with a learning rate of 
0.001
 is used to optimize the weights in the ST-GC layer. Based on the ST-GCN model, the representation extracted from the fMRI signal can not only express the variable temporal brain activity but can also express the functional dependence between brain regions. The experimental results show that ST-GCN can predict the age and gender of a subject according to the BOLD signals. Although ST-GCN can consider both the temporal and spatial resolution of the fMRI signals, they are rarely used in brain decoding [30].

## 4. fMRI-BCI Application to Psychopsychiatric Treatment

It has greatly promoted the development of fMRI-BCI that rt-fMRI has been widely used to collect, analyze, and visualize the BOLD brain signals [3]. The fMRI-BCI outputs the real-time response of the target brain area to the devices to learn the brain areas’ regulations [87]. Different from EEG-BCI, fMRI-BCI can obtain both the temporal and spatial resolution of the brain neuron activity appropriately, then perform data analysis. Moreover, fMRI-BCI can calculate the correlation coefficients between voxels in the regions of interest (ROI) of the brain, as well as identify mental and psychological states [1]. The fMRI-BCI is a closed-loop system composed of four components: participants, signal acquisition, signal analysis, and signal feedback. In the signal acquisition stage, fMRI signals of subjects’ brain activity are obtained by a magnetic resonance scanning machine. The visual signal components perform fMRI data retrieval, correction processing, and statistical analysis, and the ROI that needs to be selected on the function diagram will be labeled. Afterwards, the ROI time series output of the visual signal component is imported into the signal feedback component, which provides a real-time feedback to the subjects in a certain way [87,88]. Since fMRI-BCI is able to target the diseased cerebral cortex, it can be used to treat certain physical and physiological diseases, such as: rehabilitation, mood disorders, and attention deficit and hyperactivity disorder (ADHD). Although Hinterberger et al. have proven that visual feedback is the best option to predict brain activity patterns, the BCI performance will not be affected by the type of feedback, whether it is touch or hearing [89,90].

### 4.1. Stroke Rehabilitation

For stroke and epilepsy patients, fMRI-BCI is a potential clinical application in rehabilitation treatment to recover patients’ motor and cognitive functions [87,91]. fMRI-BCI uses neurofeedback (NF) to learn the self-regulation ability of brain regions. The fMRI-BCI will transfer to the EEG-BCI [92,93], which is more flexible and lower cost. The EEG-BCI takes the patient’s EEG as the input to recover patients’ motion ability through real-time brain ROI’s feedback [9,11,94]. In particular, Pichiorri et al. combined BCI with Motor Imagery (MI) to help stroke rehabilitation using sensorimotor BCIs [95]. This combination allow one to control external devices through direct brain activity recognition by a computer, bypassing neuromuscular-based systems (voice, use of a mouse, or keyboard). Specifically, MI evokes an event-related desynchronization (i.e., a reduction in spectral power) that occurs within a certain EEG frequency band and primarily in sensorimotor cortical regions contralateral to the imagined part of the body. EEG signals of the two groups of subjects are recorded and analyzed with power spectral density, i.e., stronger desynchronization in the alpha and beta bands. The results showed the group of the subjects with BCI supported MI had more changes in EEG sensorimotor power spectra, occurring with greater involvement of the ipsilesional hemisphere, in response to MI of the paralyzed trained hand. Through functional measurements for stroke recovery, such as high-density EEG and single-pulse transcranial magnetic stimulation (TMS), visual guidance, motion-oriented, event-related synchronization, or desynchronization feature signals are analyzed as rehabilitation exercise control signals. In addition, the connection between post-stroke cerebral hemispheres in a certain frequency band is obtained by partial directed coherence. FMRI-BCI provides patients and therapists with a means to monitor and control MI, and through rich visual feedback consistent with the image content, it can promote their adherence to brain exercises and help patients restore motor function.

### 4.2. Chronic Pain Treatment

Many patients suffering from chronic pain felt that the original pain had not disappeared even after the disease was cured. This might be attributed to the function of memory. Once the memory nerve channel is formed, it is difficult to erase. Therefore, a psychological treatment plan must be designed to cope with the pain [96]. Rostral Anterior Cingulate Cortex (rACC) is the brain area responsible for the pain perception and regulation. The subjects can thus control the rACC activation through fMRI feedback learning [97]. Through fMRI feedback training, the subjects adjust the activity of voxels in rACC, and the degree of pain will also vary with the activation of the cerebral cortex. The experiments showed that when there was no fMRI feedback training, the pain of the subjects did not decrease. Guan et al. set up the comparative experiments to verify the adjustment effect of fMRI-BCI therapy on pain [98]. They divided the subjects into a testing group and a reference group. In the testing group, the subjects adjusted the activation of rACC through neurofeedback training, and the subjects in the reference group received neurofeedback in posterior cingulate cortex (PCC). The experimental results showed that the subjects in the neurofeedback training of the rACC area had reduced pain. Therefore, it can be concluded that neurofeedback training based on fMRI-BCI may be an effective way to relieve chronic clinical pain.

### 4.3. Emotional Disorders Treatment

Whether in studying, living, or working, one must master the ability to regulate emotions and respond to setbacks and failures with a positive attitude; otherwise, emotional disorders may occur. In [99], the participants can regulate BOLD-magnitude in the right anterior insular cortex. Two different control conditions of non-specific feedback and mental imagery were set up to confirm that rtfMRI feedback is area-specific. The experiment of [100] once again verified this conclusion. Twenty-seven participants were divided into three groups, equally, and the three groups received specific feedback, non-specific feedback, and no feedback, respectively, of BOLD in the insular area. After each group received several fMRI training sessions, each group evaluated the disgusting images. The results of the experiment showed that the participants who received the BOLD feedback in the insular area had the highest scores on the disgusting images, while the other two groups did not change significantly, so the mood did not change much. The insular area plays an important role in mood regulation and may be used in the treatment of clinical emotional disorders. In addition to the insular area related to emotion regulation, it has been proved that through behavioral tests, the activity of the amygdala in the brain is closely related to the generation, recognition, and regulation of emotions and can autonomously regulate the BOLD activity in the amygdala through fMRI neurofeedback learning [101,102]. In recent years, a connectivity-neurofeedback method based on fMRI has been proposed in [103]. After neurofeedback training, participants can automatically adjust the connectivity between the amygdala and other areas of the brain to enhance their ability of emotion regulating. Unlike the aforementioned reduction in negative emotions, the research in [103] aims to enhance positive emotions. Furthermore, fMRI neurofeedback is used to treat eight patients with major depression [104], which confirmed that fMRI-based neurofeedback was an effective aid to the current treatment of depression.

### 4.4. Criminal Psychotherapy

Although more and more researchers have achieved many promising results in the treatment of mental illness [103,104], however, in modern society, patients with criminal psychology have a much higher crime rates than mental patients [105]. The current neurobiology field cannot explain the neural mechanism of this disease [87]. It is believed that regulating the emotion-related areas (such as the amygdala) may cause changes in the patient’s mental state [106,107]. The development of the new fMRI-BCI to enable criminal mental patients to self-regulate BOLD activity in their cerebral cortex requires the joint efforts of neuroscience, psychology, and computer science.

## 5. Future Directions and Challenges

The challenges to brain decoding can be summarized in three aspects: 1. the ability of the mapping model between brain activity and visual stimuli; 2. not enough matching data between visual stimuli and brain activity; 3. fMRI signals interfered by noise [108]. Although previous studies on decoding visual stimuli of brain activity have made great achievements in classification and recognition, the performance of image reconstruction needs to be improved [21]. The following sections will indicate possible solutions and future development directions for the challenges.

### 5.1. Mapping Model Capabilities

#### 5.1.1. Multi-Analysis Mode and Deep Learning

In recent years, the combination of multi-voxel analysis mode and deep learning has drawn much attention to identify brain states. By using deep neural networks, fMRI signals can be decoded to reveal brain activities. The performance of brain decoding has been further improved [30,47]. Although a decoding model based on MVPA has been proposed [65], the multi-voxel-based decoding model presents poor interpretability, especially when the decoding model uses linear kernels. In addition, this technique is susceptible to image artifacts, such as eye movement and cardiopulmonary artifacts. In addition, the speed of neuron vascular coupling, the sensitivity of BOLD activity, and the signal-to-noise ratio of fMRI signals should also be considered. In addition to the efficiency of the algorithm and the processing speed of the hardware, we should also consider the blood coupling delay in the brain [66]. The existing deep-learning-based decoding models [49] have achieved satisfactory results. Furthermore, ref. [109] correlate fMRI activity with multiple modalities, both visual and semantic features extracted from the viewed images. However, in order to obtain a higher-precision decoding model, there are still many challenges to reconstructing the corresponding visual stimuli from fMRI data by deep learning algorithms.

#### 5.1.2. ROI and Feature Selection

The sample size of the fMRI signal and image pairing is small, while the dimensionality of the fMRI signal is higher. When the model is trained with limited high-dimensional data samples, it is easy to produce dimensionality [65]. The traditional methods are in danger of overfitting on small data sets [49]. The efficiency of deep-learning-based models depends on the number and reliability of training samples. Therefore, a large number of neural activities and corresponding images can improve the quality of the reconstructed image [77]. However, the running time of the experiment should be proportional to the efficiency, so it is particularly important to select the key features that contribute most to the image reconstruction and further to improve the feature extraction ability of the decoding model for neuroimaging data [30,62]. The visual attention [110] and axiomatic attribution [111] methods in computer vision can be utilized to determine which voxels of neurons contribute most to decoding visual stimuli.

In addition, the connection of the brain network of human cognition has become one of the major interests in neuroscience research [47]. The current decoding of brain activity is usually limited to the specific cognitive areas that humans can understand, and it takes a relatively long time to collect and record fMRI signals of brain activity in these areas. Moreover, most of the current deep learning-based research cannot simultaneously consider the functional dependence and time-variant information between different regions of the brain [45]. In order to use the dependency between the ROI regions of the brain to decode the brain activity, GCN is explored to predict or annotate the cognitive state of the brain [46,47]. In particular, based on the ST-GCN model, the representation extracted from the fMRI signal contains both the temporal information of brain activity and the functional dependence between the brain regions. This method of integrating edge importance and spatio-temporal map may have potential effects on the development of neuroscience [45].

#### 5.1.3. Unsupervised Learning and Prior Knowledge

To learn an abstract representation of the brain activity in an unsupervised way should be fully studied in the future. Researchers’ exploration of the unsupervised learning methods led to the emergence of bidirectional generative models. For example, VAE is an unsupervised learning model. However, in the design of the VAE computing components, the encoder and the decoder are not related, but in the activities of the cerebral cortex, the feedforward and feedback processes are always related [50]. In addition, VAE does not have the ability to process dynamically and cyclically, but video information can be transmitted in both temporal and spatial space [50]. Reconstructing the dynamic features from brain activity is a huge challenge [21]. Some researchers used a large number of image priors to reconstruct visual stimuli [23]. When there is a priori knowledge, the decoder is a function of both brain neuron activity and prior knowledge. It is difficult to determine which information of the brain is decoded [30]. Therefore, the prior knowledge used in brain decoding still requires constant investigation of the researchers [31]. In recent years, an encoding model based on deep learning has emerged, which trains deep neural networks to perform representation learning that can predict neuronal activity [70]. Specifically, these deep learning based encoding models used visual stimuli to predict the neural response of the brain and served as a priori knowledge for the decoding model. The advancement of brain encoding has practical significance for brain enhanced communication, brain controlled machines, and disease state monitoring and diagnosis [31]. In summary, complementary model of encoding and decoding is a promising direction [30].

### 5.2. Limited fMRI and Image Data

#### 5.2.1. Few-Shot Learning

Due to the high cost of fMRI research and the complicated research process, the collection of paired fMRI signals and image samples is almost an mission impossible, so the data amount is quite small [65,77,87]. Inspired by the field of computer vision, the few-shot learning for brain activity decoding is proposed in [77], which is promising in solving the data problem of neuroimaging. There are three ways to learn few-shot: the representation-based paradigm, the initialization-based paradigm, and the illusion-based paradigm.

**The Representation-based paradigm** aims to learn the representation of fMRI signals, which regards the first layer of the neural network as a feature extractor and the last layer as a classifier. A large amount of training data is used to train the neural network to extract relevant hidden representations and complete the training of the classifier. Later, when a limited number of training examples are available, the classifier extracts a small amount of data characterization and complete the classification of the new data.

**The Initialization-based paradigm** is also called meta-learning, the idea of which is to learn how to learn. This method aims to learn good initialization parameters so that the model can cope with various new data sets. In the process of meta-learning, the previous neural network can be understood as a low-level neural network, and the meta-learner is used to optimize the weights of the low-level neural network. The meta-learner inputs a list of samples and their corresponding labels. When training the meta-learner, the meta-loss (the error between the prediction and the target label) can be used to measure the performance of the meta-learner on the target task. Then, another meta-learner is needed to update the weight of the current meta-learner.

**The Illusion-based paradigm** is to perform a series of deformation operations such as rotation or combination of the samples in the original data set to increase training examples.

#### 5.2.2. Transfer Learning

When the amount of data is limited and the prior knowledge is sufficient, sometimes the functions designed by hand are better than the neural network models learned from the data [30]. As the data amount in the fMRI data set continues to increase, manually designed functions will be replaced by the data-driven methods in the future. However, in neuroimaging, there is always a lack of data sets with large enough samples for specific experiments [112]. Most of the current transfer learning is to learn the data representation of the image in ImageNet and then build a model to adjust the medical image [73] or classify behavioral tasks from brain signals [113]. Although transfer learning can effectively make up for insufficient training data, natural images and medical images are quite different in nature [112]. For example, Gabor filters are often used for edge detection of natural images but have never been used in medical images. More and more studies have shown that the human cognitive system is a function of multiple functional areas of the brain [114]. Graph convolutional network models were trained with a a large number of medical imaging data sets in different experimental tasks and environments [112]. The experiments showed that the representation of the brain dynamics could be transferable between different ROI regions and different cognitive domains and even between different scanning sequences. Through fine-tuning, more can be learned about the high-level representation of brain functional areas, while preserving the low-level representation of the brain dynamics [47], which proved that transfer learning can not only improve decoding performance but also play a potential role in neuroimaging.

#### 5.2.3. Graph Convolutional Networks

Ref. [115] used spatiotemporal convolution neural network to jointly extract spatiotemporal features from the target network. However, if CNNs are not trained, no effective features can be obtained at all. If GCNs are not trained and use completely randomly initialized parameters, the features extracted by GCN are still effective. If labeled information is given, GCNs will perform even better [78]. Compared with other classifiers, GCNs have better performance on a limited data set [77]. The fMRI signal can represent the spatial structure of brain activity, and GCNs can consider the connectivity of the brain to perform decoding, which has the potential to solve the problem of limited data [47,77,116,117].

### 5.3. fMRI Noise

#### 5.3.1. Hemodynamic Delay

The spatial resolution of fMRI is very high, but its time resolution is relatively limited. It can only collect the average activities in about two seconds, and there is a certain delay in the detection of neural activity [3]. The fMRI signal contains the position information in the brain voxels, but due to its limited time resolution, sometimes the time series cannot be used to decode brain activity [30]. Because of the neurovascular coupling, the fMRI response is delayed after the neurological response [50]. Therefore, when fMRI signals are decoded into the latent variables of visual stimulation, the delay of the neurovascular dynamics should also be considered [50].

#### 5.3.2. Brain Cognitive Limitation

FMRI-based BCI learns the self-regulation ability of ROI in the way of neural feedback. Due to the high cost of fMRI research and the complex research process, fMRI-based BCI can be transferred to the more flexible and lower cost EEG-BCI [92,93]. VAE and GAN can be combined to use fMRI data as the supplement of the EEG data, and then encode conditional vectors with less noise [2]. In addition to decoding low-level visual stimuli, researchers [118,119,120] decode the brain activity into low-level pixel space and high-level semantic space at the same time. Due to the inadequacy of human research on visual mechanisms, the current reconstruction methods are under exploration. In the reconstruction process, the decoded noise may be the true response of the brain’s visual cortex to the outside world, while the reconstructed clear image may be the noise [59].

## 6. Conclusions

This survey investigates brain activity decoding models based on machine learning and deep learning algorithms via fMRI. The relationship between the brain activity decoding model and the brain–computer interface is closely related to the development of the brain activity decoding model and promotes the development of fMRI-BCI. Furthermore, the specific application of fMRI-BCI in the treatments of mental and psychological diseases have been investigated. Finally, it outlines the current challenges in reconstructing visual stimuli from brain activity and potential solutions. With the advancement of brain signal measurement technology, the development of more complex encoding and decoding models, and better understanding of the brain structure, “mind reading” will become true soon.

## Figures and Tables

**Figure 1 brainsci-12-00228-f001:**
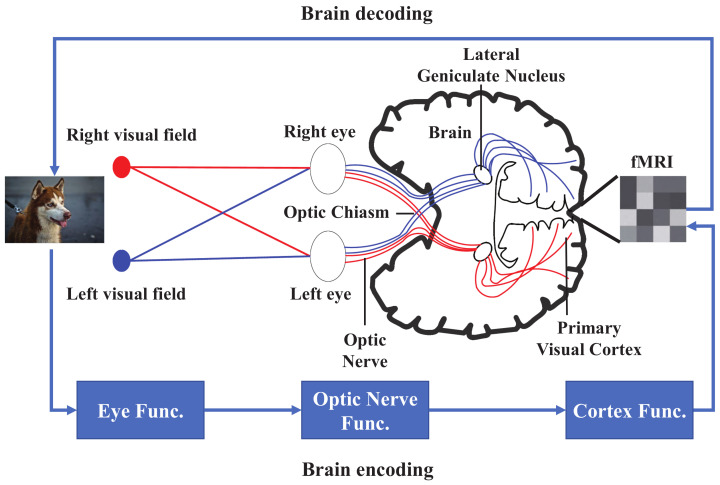
Visual pathway.

**Figure 2 brainsci-12-00228-f002:**
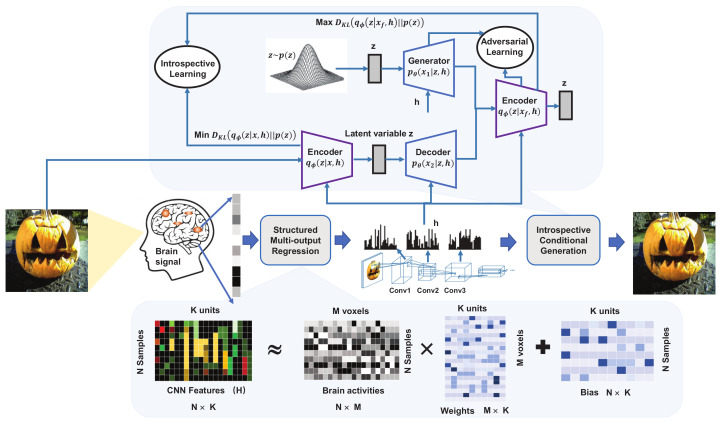
Structured deep generative neural decoding model [51].

**Table 1 brainsci-12-00228-t001:** Brain encoding models.

Literature	Objective	Model/Method	Explanation
[71]	Predict cortical responses	A pre-trained DNN	Train a nonlinear mapping from visual features to brain activity with a pre-trained DNN (i.e., AlexNet) using transfer learning technique.
[15]	Representation of information in the visual cortex	GLM	A systematic modeling method is proposed to estimate an encoding model for each voxel and then to perform decoding with the estimated encoding model.
[18]	Predict responses to a wide range of stimuli of the input images	Two-stage cascade model	This encoding model is a two-stage cascade architecture of a linear stage and nonlinear stage. The linear stage involves calculations of local filters and division normalization. The nonlinear stage involves compressive spatial summation and a second-order contrast.
[72]	Predict the response of a single voxel or brain neurons in a region of interest in any dimensional space of the stimulus	Receptive Field (rPF)	The encoding model quantifies the uncertainty of neuron parameters, rPF size, and location by estimating the covariance of the parameters.
[20]	Map the brain activity to natural scenes	Feature-weighted Receptive field	This method converts visual stimuli to corresponding visual features and assumes that spatial features are separable and uses visual feature maps to train deep neural networks. The pre-trained deep neural network weights the contribution of each feature map to voxel activity in brain regions.

**Table 2 brainsci-12-00228-t002:** Brain decoding model based on machine learning.

Literature	Objective	Model/Method	Explanation
[12]	Classification of visual stimuli decoded from brain activity	GLM	For different types of stimuli (objects and pictures), the cerebral cortex has different response patterns through the fMRI of the abdominal temporal cortex.
[27]	fMRI and brain signals classification	MVPA	For fMRI in the pre-defined ROI of the cerebral cortex, the activated mode of fMRI is classified by the multivariate statistical pattern recognition.
[64]	fMRI signals classification	SVM	SVM classifier finds the best area in the cerebral cortex that can distinguish the brain state, and then SVM is trained to predict brain state through fMRI.
[14]	Classify fMRI activity patterns	MVPA Bayesian method	The Gaussian Naive Bayes classifier has high classification accuracy. Because it assumes that the importance of each voxel is the same and it does not consider the sparsity constraint, its interpretability is poor.
[29]	Reconstruct geometric images from brain activity	MVPA	Based on a modular modeling method, the multi-voxel pattern of fMRI signals and multi-scale vision are used to reconstruct geometric image stimuli that composes of flashing checkerboard patterns.
[31]	Reconstruct the structure and semantic content of natural images	Bayesian method	Use Bayes’ theorem to combine encoding model and prior information of natural images to calculate the probability of a measured brain response due to the visual stimuli of each image. However, only a simple correlation between the reconstructed image and the training image can be established.
[16]	Improve fMRI Bayesian classifier accuracy	MVPA Bayesian method	The sparsity constraint is added to the multivariate analysis model of the Bayesian network to quantify the uncertainty of voxel features.
[23]	Reconstruct spatio-temporal stimuli using image priors	Bayesian method	This method used a large amount of videos as a priori information and combines the videos with a Bayesian decoder to reconstruct visual stimuli from fMRI signals.
[33]	Decoding human dreams	SVM	SVM classifier is trained to map natural images to brain activities, and a vocabulary database is used to label the images with semantic tags to decode the semantic content of dreams.
[22]	Decode the reversible mapping between brain activity and visual images	BCCA	The encoding and decoding network is composed of generated multi-view models. The disadvantage is that its linear structure makes the model unable to express the multi-level visual features of the image, and its spherical covariance assumption cannot understand the correlation between fMRI voxels, making it more susceptible to noise.
[34]	Dimensionality reduction of high-dimensional fMRI data	PCA	PCA reduces the dimensionality of the facial training data set, and the partial least squares regression algorithm maps the fMRI activity pattern to the dimensionality-reduced facial features.
[24]	Infer the semantic category of the reconstructed image	Bayesian method	Propose a mixed Bayesian network based on the Gaussian mixture model. The Gaussian mixture model represents the prior distribution of the image and can infer high-order semantic categories from low-order image features through combining the prior distributions of different information sources.
[28]	Predict object categories in dreams	MVPA CNN	Based on CNN, train a decoder with the data set of the normal visual perceptions, and decode the neural activity to the object category. This process involves two parts: 1. map the fMRI signal to the feature space; 2. using correlation analysis to infer the object category based on the feature space.
[44]	Decode visual stimuli from human brain activity	RNN CNN	Use CNN to select a set of small fMRI voxel signals as the input and then use RNN to classify the selected fMRI voxels.
[41]	Capture the direct mapping between brain activity and perception	CNN	The generator is directly trained with fMRI data by an end-to-end approach.
[40]	Reconstruct dynamic video stimuli	CNN PCA	The CNN-based coding model extracts the linear combination of the input video features and then uses PCA to reduce the dimensionality of the extracted high-dimensional feature space while retaining the variance of 99% of the principal components.

**Table 3 brainsci-12-00228-t003:** Deep generative models based on VAE or GAN.

Literature	Objective	Model/Method	Explanation
[21]	Reconstruct perception images from brain activity	Deep Generative Multiview Model (DGMM)	DGMM first uses DNN to extract the image’s hierarchical features. Based on the fact that the human brain’s processing model for external stimuli is sparse, a sparse linear model is used to avoid over-fitting of fMRI data. The statistical relationships between the visual stimuli and the evoked fMRI data are modeled by using two view-specific generators with a shared latent space to obtain multiple correspondences between fMRI voxel patterns and image pixel patterns. DGMM can be optimized with an automatically encoded Bayesian model [32,75].
[37]	Reconstruct facial images	Deep Adversarial Neural Decoding (DAND)	DAND uses the maximum posterior estimation to transform brain activity linearly to the hidden features. Then, the pre-trained CNN and adversarial training are used to transform the hidden features nonlinearly to reconstruct human facial images. DAND showed good performance in reconstructing the details of the face’s gender, skin color, and facial expressions.
[48]	Improve the quality of reconstructed images	Introspective Variational Autoencoders (IntroVAE)	IntroVAE generator and inference model can be jointly trained in a self-assessment manner. The generator takes the output of the inference model noise as the input to generate the image. The inference model not only learns the potential popular structure of the input image but also classifies the real image and the generative image, which is similar to GAN’s adversarial learning.
[59]	Reconstruct natural images from brain activity	Deep Convolution Generative Adversarial Network (DCGAN)	DCGAN uses a large natural image data set to train a deep convolutional generation confrontation network in an unsupervised manner, and learn the potential space of stimuli. This DCGAN is used to generate arbitrary images from the stimulus domain.
[60]	Reconstruct the visual stimuli of brain activity	GAN	They used an encoding model to create surrogate brain activity samples, with which the generative adversarial networks (GANs) are trained to learn a generative model of images and then generalized to real fRMI data measured during the perception of images. The basic outline of the stimuli can finally be reconstructed.
[50]	Reconstruct visual stimuli (video) of brain activity	VAE	VAE is trained with a five-layer encoder and a five-layer decoder to learn visual representations from a diverse set of unlabeled images in an unsupervised way. VAE first converts the fMRI activity to the latent variables and then converts the latent variables to the reconstructed video frames through the VAE’s decoder. However, VAE could only provide relatively lower accuracy in higher-order visual areas compared to CNN.
[38]	Reconstruct color images and simple gray-scale images	GAN	A pre-trained DNN decodes the measured fMRI patterns into the hierarchical features that can represent the human visual layering mechanism. The DNN network extracts image features, and then compares them with the decoded human brain activity features, which guides the deep generator network (DGN) to reconstruct images and iteratively minimizes the errors of the two. A natural image prior introduced by an enhanced DGN semantically details to the reconstructions, which improves the visual quality of generated images.
[51]	Reconstruct the visual image from brain activity	A structured multi-output regression (SMR) model and Introspective Conditional Generation (ICG)	Decodes the brain activity to the intermediate CNN features and then maps these intermediate features to visual images. Combining maximum likelihood estimation and adversarial learning, ICG model uses divergence and reconstruction error for adversarial optimization, which can evaluate the difference between the generated image and the real image.
[76]	Use semantic features to add details to the generated image	Shape-Semantic GAN	This framework consists of a linear shape decoder, a semantic decoder based on DNN, and an image generator based on GAN. The output of the shape decoder and the semantic decoder are input to the GAN-based image generator, and the semantic features in GAN are used as a supplement to the image details to reconstruct high quality images.
[57]	Reconstruct natural images from brain activity	Progressively Growing GAN (PG-GAN)	This model adds a priori knowledge of potential features to (PG-GAN). The decoder decodes the measured response of the cerebral cortex into the latent features of the natural image and then reconstruct the natural image through the generator.
[58]	Reconstruct natural images from brain activity	Similarity-conditions generative adversarial network (SC-GAN)	SC-GAN not only extracts the response patterns of the cerebral cortex to natural images but also captures the high-level semantic features of natural images. The captured semantic features is input to GAN to reconstruct natural images.

**Table 4 brainsci-12-00228-t004:** GCN-based brain decoding.

Literature	Objective	Model/Method	Explanation
[45]	Localize brain regions and functional connections	Spatio-Temporal Graph Convolution Networks (ST-GCN)	Based on ST-GCN, the representation extracted from the fMRI data expresses both temporal dynamic information of brain activity and functional dependence between brain regions. Through training ST-GCN, this method can learn the edge importance matrix on short sub-sequences of BOLD time series to improve the prediction accuracy and interpretability of the model.
[46]	Decode the consciousness level from cortical activity recording	BrainNetCNN	BrainNetCNN is a GCN-based decoding model, which uses multi-layer non-linear units to extract features and predict brain consciousness states.
[47]	Predict human brain cognitive state	GCN	The brain annotation model uses six graph convolutional layers as feature extractors and two fully connected layers as classifiers to decode the cognitive state of the brain, taking a short series of fMRI data as the input, spreading the information in the annotation model network, and generating high-level domain-specific graph representations to predict the brain cognitive state.

## Data Availability

Not applicable.

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
