# Peer review of "fMRI Brain Decoding and Its Applications in Brain–Computer Interface: A Survey"

_brainsci, 2022, doi:10.3390/brainsci12020228_

Round 1

Reviewer 1 Report

This review paper presents an interesting topic for the audience of Brain Sciences. The manuscript is easy readable and organized, but it still can be benefited after proofreading. In the manuscript the authors should describe how the literature review was conducted for other readers to replicate the review methodology. Furthermore, the scientific issues and the hypothesis can be described, as well as what search strings or keywords were used, what datasets were queried, what years were included, inclusion/exclusion criteria. The references should be updated, as the current manuscript has a total of 121 references, being 32.23% published before 2012, while 60.33% of the studies were published before 2017.

Minor concerns: Check the sentences on the lines 297-298 pp.9, and lines 398-399 pp. 13.

Reviewer 2 Report

Please find my comments in the attached document.

Round 2

Reviewer 1 Report

The authors have attended partially my concerns. Some important aspects should be still added in section Introduction, such as such as scientific problem and and hypothesis that motivated this review. The objective of this paper (see lines 74-77 pp. 2) should be better elaborated, as other topics are further discussed in sections 4. fMRI-BCI application to psychopsychiatric treatment,  4.1. Stroke rehabilitation, 4.2. Chronic pain treatment, and 4.3. Emotional disorders treatment.

Before the last paragraph in section Introduction, also explain how this review was conducted, describing what search strings or keywords were used, what datasets were queried, what years were included, inclusion/exclusion criteria. 

Please check if 4.1. Stroke rehabilitation should be included in 4. fMRI-BCI application to psychopsychiatric treatment, as these two topics sound different.

Reviewer 2 Report

The authors have addressed some of my previous comments and concerns. I think the work looks better now. However, there are still some misleading details and copyright issues that have to be addressed. Also, most responses are based on arguments that are not convincing. Here are some examples and comments.

Regarding comment 2. I think there is a copyright issue here, and simply changing the title does not resolve the problem. For instance, it looks like Figure 2 is a summary of the work by Du et al. 2020. I wonder if the authors have requested permission to include their figures in this submission. If so, it has to be specified in the figure caption with appropriate citations. This applies to all the presented figures.

Regarding comment 4, "The photon frequency of fMRI radiation is very low, much lower than sunlight, and does not interfere with various processes in the human body, and the total radiation power absorbed by the body is also very low, so the thermal effect can also be ignored." is not a scientific argument because no references have been provided. Also, the claim that "besides, compared with other physiological signals such as electroencephalography (EEG), fMRI has more potential for clinical applications" is wrong. It depends on the application. EEG is commonly (almost always) used for epilepsy and long-term monitoring, whereas fMRI is used for mapping functional activities. Also, fMRI is indeed an indirect measure (blood flow) of neural activity.

Comment 5. The suggested (paragraph) is missing in the revision/manuscript, and I don't think citation 92 belongs to the OPM-MEG original work. I would like to mention that the OPM-MEG is simply not fMRI and so incomparable. 
